# Maternal Supplementary Tapioca Polysaccharide Iron Improves the Growth Performance of Piglets by Regulating the Active Components of Colostrum and Cord Blood

**DOI:** 10.3390/ani13152492

**Published:** 2023-08-02

**Authors:** Shengting Deng, Chengkun Fang, Ruiwen Zhuo, Qian Jiang, Yating Song, Kaili Yang, Sha Zhang, Juanyi Hao, Rejun Fang

**Affiliations:** 1College of Animal Science and Technology, Hunan Agricultural University, Changsha 410128, China; dengshengting97@163.com (S.D.); clark_fang@163.com (C.F.); zhuoruiwen97@163.com (R.Z.); jiangqian@hunau.edu.cn (Q.J.); syt3523622753@126.com (Y.S.); kellyyang94@126.com (K.Y.); zs230108@126.com (S.Z.); h972751748@126.com (J.H.); 2Hunan Engineering Research Center of Intelligent Animal Husbandry, Changsha 410128, China

**Keywords:** sow, polysaccharide iron, piglet, colostrum, cord blood

## Abstract

**Simple Summary:**

Nowadays, iron complexation of polysaccharides has recently become a popular topic in glycobiology research. A macromolecular polysaccharide known as polysaccharide–iron complex can affect blood pressure, blood sugar levels, immunity, blood circulation, and has the potential to improve and regulate active substances that exist in animal blood. In this study, an organic iron complex generated from tapioca starch and ferric chloride was utilized. This product is a self-developed, plant-derived, natural compound that has a wide variety of sources and is cost-effective. The results of this study demonstrate that maternal supplementary tapioca polysaccharide iron can improve their feed intake and reproductive performance, adjust the nutritional composition of colostrum, enhance the antioxidant and immunological performance of piglets, and stimulate hormone release in blood, leading to enhanced piglet production performance. Furthermore, this study reveals the correlation between the components of colostrum, cord blood, and piglet performance when sows are fed supplements with tapioca polysaccharide iron.

**Abstract:**

The purpose of this study was to investigate the effect of maternal supplementation with TpFe (tapioca polysaccharide iron) on reproductive performance, colostrum composition, cord blood active components of sows, and growth performance of their nursing piglets. Sixty healthy Duroc × Landrace × Yorkshire sows were randomly assigned to three groups at day 85 of gestation. The experimental diets included a basal diet supplemented with 100 mg/kg FeSO_4_·H_2_O (CON group), the basal diet supplemented with 50 mg/kg TpFe (TpFe50 group), and the basal diet supplemented with 100 mg/kg TpFe (TpFe100 group), as calculated by Fe content. The experiment lasted from day 85 of gestation to the end of weaning (day 21 of lactation). Results showed that maternal supplementation with 100 mg/kg TpFe improved (*p* < 0.05) feed intake during lactation, live births, and birth weight of the litter (alive) and increased (*p* < 0.05) colostrum IgM (immunoglobulin m), IgA (immunoglobulin A), as well as the IgG levels, while it decreased (*p* < 0.05) the urea nitrogen and somatic cell count of sows. Moreover, sows in the TpFe100 group had higher (*p* < 0.05) serum iron levels and IgG. Additionally, maternal supplementation with 100 mg/kg TpFe increased (*p* < 0.05) iron level, total antioxidant capacity (T-AOC), glutathione peroxidase (GSH-px), catalase (CAT), IgG, red blood cells (RBC), and hemoglobin (Hb) of cord blood, similar with the iron content, T-AOC, GSH-px, IgG, RBC, Hb, hematocrit (HCT), and mean corpuscular volume (MCV) of weaned piglet blood. The diarrhea and mortality rates among the nursing piglets were decreased (*p* < 0.05), while the average weight at day 21 of age was increased (*p* < 0.05) in the TpFe100 group. Serum PRL (prolactin) levels of sows exhibited a positive correlation (*p* < 0.05) with live births. Suckling piglet diarrhea was positively correlated with colostrum urea nitrogen level but negatively correlated with colostrum IgM, IgG, and cord blood Hb content (*p* < 0.05). The mortality of suckling piglets was negatively correlated with serum iron content and IgM in colostrum, GSH-px, and IgG in cord serum of sows (*p* < 0.05). The average weight of weaning piglets was positively (*p* < 0.05) related to colostrum IgM and IgG levels, as well as cord serum RBC counts of sows on day 21. In conclusion, maternal supplementation with TpFe can improve the active components of colostrum and umbilical cord blood and improve the growth performance of suckling piglets.

## 1. Introduction

Iron is a crucial nutrient for animal growth, reproduction, and immunity. Hemoglobin (Hb), a component of red blood cells (RBCs), requires iron to facilitate the physiological process of oxygen transfer [1]. Iron deficiency in pregnant animals can lead to decreased litter weight, increase in stillbirths, and neonatal infections [2]. Furthermore, iron deficiency is often related to iron-deficiency anemia (IDA), which has negative health consequences and frequently results in early mortality in piglets. Due to their unique iron properties, newborn piglets are particularly vulnerable to IDA. Compared to other animals, newborn piglets have insufficient iron storage, increased iron consumption, and a lack of iron supply in breast milk [3]. Iron supplements are typically given to animal diets to supplement iron elements, with inorganic iron supplements (ferrous sulfate) being the most commonly utilized. However, inorganic iron supplements are known to have low bioavailability, produce harmful side effects, and pollute the environment [4]. Numerous studies on the efficacy of iron supplementation in pregnant sows have demonstrated that organic iron supplements, such as ferrous fumarate and amino-acid-chelated iron, have been confirmed to perform better than inorganic iron supplements in animal production applications [1,5,6]. Iron storage in piglets increased significantly with the level of dietary glycine chelated and methionine iron during the early pregnancy and suckling periods [7], and iron content in breast milk also increased significantly [8]. However, it has also been reported that inconsistent results have been obtained, which may be related to the complexity of the existing organic iron processing technology and poor product stability [9]. Therefore, it is necessary to find a new and more efficient iron supplement.

Nowadays, iron complexation of polysaccharides has recently become a popular topic in glycobiology research. Saccharide–iron complexes as novel iron supplements have aroused attention for the high iron absorption rate and no gastrointestinal irritation in oral doses. In addition, research on the biological activities of saccharide–iron complexes revealed that they also exhibited good abilities in treating anemia, eliminating free radicals, and regulating the immune response [10,11]. In this study, an organic iron complex generated from tapioca starch and ferric chloride was utilized. This product is a self-developed, plant-derived, natural compound that has a wide variety of sources and is cost-effective. The substantial potential of polysaccharide iron compounds has been reported [10,12,13]. However, there has been no research on the effect of TpFe in iron supplementation using pig models. The National Research Council (NRC) recommends a minimum iron requirement of 80 mg/kg for late gestation and nursing sows [14]. However, due to unidentified factors such as feed processing waste, it has been demonstrated that sows benefit from an exogenous iron supplement dose of around 100 mg/kg [3,15]. Thus, in this experiment, a half-dosage group of 50 mg/kg iron dextran was set up to determine if the half-dose could replace 100 mg/kg inorganic iron (ferrous sulfate monohydrate), as well as the benefits of 100 mg/kg iron dextran in the same dose group. In this study, we investigated the effects of maternal TpFe supplementation on sow reproductive performance, colostrum and cord blood active components, and growth performance of piglets, and for the first time determined the correlation between maternal dietary TpFe management and offspring growth performance.

## 2. Materials and Methods

### 2.1. FeSO_4_·H_2_O and TpFe Samples

FeSO_4_·H_2_O, a feed-grade inorganic iron supplement, was obtained from Chengdu Shuxing Feed Co., Ltd., Chengdu, China, with an iron content of 29.0% in the samples. TpFe was produced by Guangxi Research Institute of Chemical Industry (Nanning, China). Briefly, tapioca starch was degraded with acids, underwent alkaline reactions, complexed (supplemented) with ferric chloride and sodium hydroxide, and then had salts and impurities removed with nanofiltration. The resulting TpFe was spray-dried and used in the experiment, with the iron content of the samples being 30.0%.

### 2.2. Animals, Diets, and Experimental Design

The current experiment adopted a single-factor variable design. Sixty healthy Duroc × Landrace × Yorkshire sows (with an average parity of 2.19 ± 0.15 and an average BW of 236.65 ± 5.40 kg) with a gestation age of 85 days were randomly divided into three treatment groups. The experimental diets included a basal diet supplemented with 100 mg/kg FeSO_4_·H_2_O (CON group, *n* = 20), the basic diet supplemented with 50 mg/kg TpFe (TpFe50 group, *n* = 20), and the basal diet supplemented with 100 mg/kg TpFe (TpFe100 group, *n* = 20). The supplemental levels were calculated based on the iron content. The preparation of experimental feed to meet the sows’ nutritional requirements (NRC, 2012), including the ingredients and nutrients, is presented in Table 1. The experiment was conducted from day 85 of gestation to day 21 of lactation (end of weaning). Pregnant sows were housed separately in a 2.4 m^2^ limit stall until day 108 of pregnancy, and then in a 1.8 m^2^ farrowing limit stall until the experiment was completed. Both gestation and lactation diets were in pellet form. Initially, the premix in the basic diet was divided into three equal parts and weighed. Then, based on the amount of iron added to each treatment (dextran iron or ferrous sulfate monohydrate), each equal part of the premix was mixed in a 100 kg small high-speed mixer. Finally, the three premixes and the remaining treated base diet aniseed were thoroughly mixed in a large horizontal banded blade mixer and then granulated. Sows were fed a 3.0 kg diet per day, which was split into two meals: half at 07:00 and half at 15:00, starting from day 85 of gestation until farrowing. During lactation, the feed was offered three times a day at 07:00, 15:00, and 21:00, and all sows were allowed to consume diets ad libitum. Sows and piglets had free access to drinking water. Additionally, data from sows with illnesses, serious lameness, deaths, and reproductive failure were excluded from further analysis. On the third day after birth, 200 mg of iron dextran was administered to each piglet.

### 2.3. Recording and Sample Collection

The feed intake of sows during pregnancy and lactation, as well as the numbers of litter size, live births, stillbirths, mummies, duration of labor, birth (alive) litter weight, and average weight of live-born piglets were recorded. Cross-fostering was kept within diet treatments to adjust litter size within 48 h postpartum to ensure that the litter size of each sow in the same treatment group is similar. Feed samples were collected and stored at −20 °C until analysis. On day 110 of gestation, blood samples totaling 10 mL were taken from the auricular veins of 9 randomly selected sows from each group. On the day of delivery, umbilical cord blood and colostrum were collected from the 1st, 2nd, 3rd, 10th, 11th, 12th, 18th, 19th, and 20th sows in each group, according to the order of sow deliveries. Nine sows were selected, and each had 10 mL of blood collected. Blood samples of 3 mL were allowed to stand at room temperature for 30 min, and the content of active substances in whole blood was determined within 4 h. The remaining 7 mL blood samples were centrifuged at 3000× *g* for 15 min at 4 °C and then separated into 1.5 mL sterile and enzyme-free tubes. They were stored in a refrigerator at −80 °C for further analysis. Colostrum was collected from each sow using 10 mL cryo-storage tubes and stored at −80 °C until later analysis. The body weight (BW) of piglets was recorded on the day of birth and at weaning on day 21. The fecal status of piglets was observed and recorded daily during lactation. The rate of diarrhea was calculated based on the scoring criteria described in a previous study [16], and the mortality rate of piglets was also recorded.

### 2.4. Chemical Analyses

The gross energy in feed and feces was determined by an SDACM3100 automatic calorimeter following the international standard ISO9831-1998 method [17]. The crude protein content was determined according to GB/T 6432-2018. Calcium content was determined according to GB/T 6436-2018, while total phosphorus content was determined using the GB/T 6437-2018 method. Iron content in the diet was determined using an atomic absorption spectrophotometer (Vario 6, Jena, Germany) according to the GB/T13885-2017 method. In brief, the sample was ashed at (550 ± 15) °C, while the residue was dissolved with hydrochloric acid and diluted to a constant volume, and then imported into the air-acetylene flame of the atomic absorption spectrometer. The absorbance of iron was measured, and the content of iron in the solution to be measured was calculated according to the concentration of the standard solution. The iron content of serum and colostrum, malonaldehyde (MDA), T-AOC, GSH-px, total superoxide dismutase (T-SOD), CAT, as well as the levels of IgA, immunoglobulin M(IgM), immunoglobulin G(IgG) in colostrum, interleukin-2 (IL-2), interleukin-6 (IL-6), tumor necrosis factor (TNF-α), CEE, PRL, cholecystokinin (CCK), and growth hormone (GH) were determined using ELISA kit (Nanjing Jiancheng Bioengineering Institute, Nanjing, China) according to the manufacturer’s instructions.

The routine active substances in blood were detected using an Auto Hematology Analyzer (Mindray BC-5000 Vet, Shenzhen, China). The determination indexes included IgA, immunoglobulin M (IgM), immunoglobulin G (IgG) in serum, RBC, Hb, HCT, and MCV. The milk protein, lactose, milk fat, fat-removed dry matter, and total dry matter contents were determined using the FOSS milk composition analyzer (MilkoScanTM FT200 Type76150, Copenhagen, Denmark), while somatic cell count was determined using a somatic cell analyzer (Fosss FC Type 79910, Copenhagen, Denmark).

### 2.5. Statistical Analysis 

A single-factor variable design was adopted in the current experiment. The level of piglet diarrhea and mortality were analyzed using the chi-square test and presented as bar graphs. Other data were analyzed by one-way ANOVA using SPSS 26.0 (SPSS. Inc., Chicago, IL, USA). Tukey’s multiple-range test was used to analyze differences, and when overall differences were significant, Duncan’s multiple-range test (SPSS 26.0) was used to test the differences. The results are presented as mean values and standard error (SEM). Hormone levels in serum analysis and performance of suckling piglets were analyzed using GraphPad Prism 9 software (v9.5.1, San Diego, CA, USA) and presented as bar graphs. Pearson’s correlation analysis was performed to evaluate potential links between indicators of significant changes in sow colostrum, serum active components, cord blood, piglet serum, and performance of newborn piglets using SPSS 26.0. Correlation analysis heatmaps were generated using GraphPad prism 9. Differences between mean values were considered statistically significant at *p* < 0.05.

## 3. Results

### 3.1. Feed Intake of Sows

The feed intake of sows during gestation and lactation were presented in Table 2. The feed intake of sows from all treatments during gestation was similar (*p* > 0.05). However, during lactation, the feed intake of sows in the TpFe100 group was increased 8.58% (*p* < 0.05) when compared to CON groups. No significant differences were observed between the CON and TpFe50 groups (*p* > 0.05).

### 3.2. Sow Reproductive Performance

The reproductive performance of the sows was summarized in Table 3. There were no significant differences in litter size, stillbirths, mummies, and duration of labor among all treatment groups (*p* > 0.05). However, the numbers of live-born piglets in the TpFe100 group were increased by 19.49% when compared to the sows of the CON group (*p* < 0.05). Furthermore, the birth (alive) litter weight and average weight of live-born piglets were increased in the TpFe100 group (*p* < 0.05) compared to the CON and TpFe50 groups. No significant differences were observed between the CON and TpFe50 groups for the number of live-born piglets, birth (alive) litter weight, and average weight of live-born piglets (*p* > 0.05).

### 3.3. Active Components of Colostrum

There were no significant affects (*p* > 0.05) in total dry matter, milk fat percentage, milk protein percentage, milk lactose percentage, non-milk fat solid content, and iron content in the colostrum of sows among the three groups (Table 4). However, sows in the TpFe 100 group had lower somatic cell counts and urea nitrogen content in their colostrum (*p* < 0.05). Moreover, the colostrum of sows in the TpFe100 group increased the levels of IgM, IgA, and IgG compared to the CON group (*p* < 0.001). Meanwhile, compared to the CON group, the colostrum of sows in the TpFe50 group improved the levels of IgM (*p* < 0.05), but no significant differences were observed in IgA and IgG (*p* > 0.05).

### 3.4. Iron Content of Diet and Serum

During late gestation and lactation, the iron content of diet in the CON and TpFe100 groups was significantly higher (*p* < 0.05) compared to the TpFe50 group (Table 5). The iron content of sow serum, cord blood, and weaned piglet blood in the TpFe100 group was increased (*p* < 0.05) when compared to the CON groups. Additionally, there were no significant differences in the iron content of sow serum, cord blood, and weaned piglet blood between the CON and TpFe50 groups (*p* > 0.05).

### 3.5. Analysis of Serum Antioxidant Capacity

Supplemental TpFe had no effects on MDA, T-AOC, GSH-px, T-SOD, and CAT levels in sow serum when compared to the CON group (*p* > 0.05) (Table 6). Interestingly, the T-AOC, GSH-px, and CAT levels in cord serum and piglet serum in the TpFe100 group were increased compared to the CON group (*p* < 0.05). Additionally, compared to the CON group, the levels of T-AOC and CAT in cord serum were increased (*p* < 0.05) in TpFe50 and TpFe100 group, whereas no significant differences were found for MDA, GSH-px, and T-SOD (*p* > 0.05) in the TpFe50 group. Similarly, the levels of CAT in piglet serum in the TpFe50 group was increased (*p* < 0.05) when compared to CON group, while there were no significant differences in MDA, T-AOC, GSH-px, and T-SOD between the two groups (*p* > 0.05).

### 3.6. Immune and Conventional Substance Active Ingredient Content of Blood

There was a significant increase (*p* < 0.05) in IgG levels in sow serum in the TpFe100 group when compared to the CON group (Table 7). However, the results for other active components of sow serum had no effects among the three treatment groups (*p* > 0.05). Compared to the CON group, the levels of IgG, RBC, and Hb in cord serum were significantly increased in the TpFe100 group (*p* < 0.05), although no significant differences were observed for other active components (*p* > 0.05). Additionally, the levels of IgG, RBC, Hb, HCT, and MCV in piglet serum were significantly higher in the TpFe100 group when compared to the CON group (*p* < 0.05). 

### 3.7. Hormone Level of Serum

Maternal supplementary TpFe did not affect the levels of IL-6, TNF-α, or CEE in sow serum (Figure 1A,B). However, the levels of IL-2, PRL, and CCK in sow serum were increased in the TpFe50 and TpFe100 groups when compared to the CON group (*p* < 0.05; Figure 1A,B). There were no significant differences in the levels of IL-2, IL-6, TNF-α, or CEE in cord blood serum among the three groups (*p* > 0.05; Figure 1C–E), but the levels of PRL and GH were increased (*p* < 0.05) in TpFe50 and TpFe100 cord blood. No significant differences were observed in the levels of IL-2, IL-6, and TNF-α among the three groups in piglet serum (*p* > 0.05; Figure 1F,G), while the level of GH in piglet serum was increased (*p* < 0.05) in the TpFe50 and TpFe100 groups compared to the CON group.

### 3.8. Performance of Suckling Piglets

The effects of TpFe supplementation in the sow’s diet on the performance of suckling piglets are presented in Figure 2. The diarrhea and mortality rates of the TpFe100 group were significantly lower than the CON group (*p* < 0.05; Figure 2A). However, there was no significant difference in litter weight between the CON and TpFe100 groups (*p* > 0.05; Figure 2B), although the average weight of weaned piglets at day 21 was improved in the TpFe100 group compared to the CON group (*p* < 0.05; Figure 2B). Additionally, the diarrhea rate also saw significant decreases in the TpFe50 group when compared to the CON group, (*p* < 0.05; Figure 2A); there were no significant differences in mortality, litter weight, or average weight of weaned piglets (*p* > 0.05; Figure 2B).

### 3.9. Correlation between Indicators of Colostrum, Cord Serum, and Performance of Piglets

A Pearson’s correlation analysis was performed to evaluate the potential links between indicators of significant changes in the sow’s colostrum, serum active components, cord blood, piglet serum, and the performance of newborn piglets (Figure 3). Live-born piglets were showed a positive correlation (*p* < 0.05, r = 0.473) with the serum PRL levels of sows (Figure 3A). Similarly, the diarrhea rate of suckling piglets also saw positive correlations (*p* < 0.05) with the urea nitrogen content in colostrum (r = 0.426), while a negative correlation (*p* < 0.05) was observed between the colostrum IgM (r = −0.385), IgG (r = −0.586), and cord blood Hb content (r = −0.586) (Figure 3B). Moreover, the mortality of suckling piglets displayed a negative correlation (*p* < 0.05) with the serum iron content (r = −0.513) of sows and IgM in colostrum (r = −0.608) (Figure 3A), as well as GSH-px (r = −0.532) and IgG of cord serum (r = −0.605) (Figure 3B). However, the average weight of weaned piglets at day 21 was positive (*p* < 0.05) with the IgM (r = 0.396) and IgG (r = 0.379) of colostrum (Figure 3A), as well as RBC (r = 0.379) of cord serum (Figure 3B).

## 4. Discussion

The use of inorganic iron in animal diets has various negative impacts in production, including gastrointestinal side effects, iron poisoning, and environmental pollution. Although several organic irons, such as ferrous fumarate and amino-acid-chelated iron, have superior effects and fewer side effects than inorganic iron, product stability is poor, and the complex synthesis process results in high production costs. TpFe, the iron preparation employed in this experiment, is a novel form of iron preparation with good stability, low cost, and independent innovation patents. In this study, we focused on the effects of maternal iron polysaccharide supplementation on the active components of colostrum and cord blood, as well as the correlation between these effects and offspring growth performance.

Inadequate feed consumption during lactation can reduce the number of nutrients available in milk production, which can restrict the growth and development of piglets [18]. Increased feed intake by sows provides greater energy and nutrients for milk synthesis, leading to improved piglet development [19,20]. In the current study, the feed intake of sows during pregnancy did not change significantly, possibly because the sows did not have ad libitum access to feed during pregnancy, but rather a limit was placed on the amount they could consume. However, the feed intake of sows in the TpFe100 group was significantly increased during lactation. This is similar to previous studies in which adding organic iron to a typical lactation diet and feeding it to sows for the 26 days prior to farrowing increased their feed consumption [21]. As we know, sow milk production increases with feed consumption during lactation, indicating that maternal TpFe supplementation provides a strong foundation for improving piglet development performance.

The effects of several organic iron chelates on sows are inconsistently described in the available literature. For example, the bioavailability of iron and the proportions of stillborn and mummified fetuses per litter were considerably enhanced by adding glycine chelate (62.5 mg/kg) to sow diets for 0, 2, 4, 6, or 8 weeks before farrowing [9]. In contrast, sows administered an organic iron complex (120 mg/kg, Fe) throughout the entire pregnancy period had larger litters [6]. However, when an organic iron chelate complex (80 mg/kg, Fe), consisting of 35% ferrous fumarate, 25% iron lactate, 37% glycine chelate iron, and 3% iron methionine chelate, was fed to sows from day 84 to parturition, the number of stillborn and mummified fetuses per litter was not reduced [22]. In our study, TpFe (50 or 100 mg/kg, Fe) was used to supplement the sows’ feed from day 85 to parturition. The litter size, number of stillborn piglets, mummies, and duration of labor were not affected by TpFe administration, but a significant improvement in the number of live-born piglets, birth (alive) litter weight, and average weight of live-born piglets was observed under the 100 mg/kg Fe in TpFe treatment. This is similar to the effects of feeding sows a diet with 80 mg Fe/kg ferrous chelate, as reported by Wan et al. [2]. These findings suggest that different reproductive effects in sows are influenced by the organic iron chelate source or structure, added concentration, and duration. However, the benefits of TpFe in improving sow reproductive performance are reflected in its ability to support fetal survival.

Colostrum synthesis begins considerably before parturition, and thus other mineral requirements in sow reproduction may impact its composition, particularly during the latter stages of fetal development during pregnancy [23]. Previous studies have reported that changes in the chemical composition of colostrum, such as fat, protein, lactose, and non-milk fat solids, also reflect different physiological conditions of lactating sows [24,25]. In our findings, the total dry matter content, milk fat percentage, milk protein percentage, milk lactose percentage, and non-milk fat solid content of colostrum were not impacted, which is similar to a previous study [26]. This discrepancy can be explained by the fact that the sow’s physical state and perpetual feeding strategy affect the nutritional content of colostrum [27]. Furthermore, leukocytes and epithelial cells make up the majority of somatic cell types in milk and colostrum. According to Maurer et al. [28], the somatic cell count (SCC) of milk is frequently used as a barometer for the well-being and quality of lactating animals, and the urea nitrogen concentration is a measure of urine nitrogen excretion that is connected to dietary crude protein (CP) intake and the ratio of degradable to undegradable protein [29]. In this study, we found lower somatic cell counts and urea nitrogen concentrations in colostrum, which may indicate that maternal TpFe supplementation improved the quality of sow colostrum and had greater bioavailability than inorganic iron.

Moreover, colostrum and milk contain different amounts of iron, and the amount of iron absorbed by piglets varies from sow to sow. Piglets consume 0.5 to 1 L of milk per day. Piglets absorb 60–90% of lactoferrin in breast milk at concentrations of 0.2–4 mg/L [30]. The iron content in colostrum can be increased by supplementing it with maternal iron chelates [31] or lactoferrin [3]. In contrast, the iron content of colostrum and breast milk remained constant at 0.15% with or without chelated iron [32]. This is consistent with our current findings of no significant difference in colostrum iron content, possibly due to the short time interval between late gestation and parturient TpFe supplementation in sows.

Immunoglobulins from the colostrum are critical for the survival and development of piglets. Infants lack globulins and therefore rely on colostrum as their primary supply of antibodies. The primary source of innate immunity against bacterial infection is mainly immunoglobulins [33]. Ma et al. [34] reported that sows fed low-dose organic iron had no significant effect on colostrum IgM, IgA, and IgG. However, in our study, dietary supplementation of 100 mg/kg in the form of TpFe significantly increased the levels of IgM, IgA, and IgG in colostrum. This may indicate that TpFe is involved in the synthesis of colostrum immunoglobulins, which also improves the survival of newborn piglets.

Serum iron, or iron bound to transferrin in serum, has been used to qualitatively assess the bioavailability of iron supplements [35]. Piglet serum iron significantly increased when Fe-Gly was added to sows’ diets, while FeSO_4_·H_2_O supplementation had some positive benefits, but they were not significant [15]. High-iron diets during pregnancy have been linked to an increase in serum iron, according to Spruill et al. [36], while iron from an amino acid complex has been linked to an increase in blood serum iron, according to Yu et al. [37]. Moreover, a previous study reported that organic iron complexes increased sow serum iron concentration on day 1 of lactation compared to ferrous sulfate [20]. In contrast, treatment with 100 mg/kg of TpFe significantly increased serum iron content in the current study. Dietary supplementation with 50 mg/kg of TpFe had no significant effect on serum iron content in sows, umbilical cord serum, or weanling piglets compared to the inorganic iron supplementation group. Surprisingly, the 100 mg/kg TpFe supplement group showed a consistent pattern of iron content changes in sow serum, cord blood serum, and weanling pig serum. This may be due to the fact that, like lactoferrin and heme iron [38], TpFe has greater potential than inorganic iron to cross the placental barrier and enter the embryo. Unfortunately, we did not focus on the expression of genes and proteins involved in iron transport in sow placentas in this work, while additional in-depth mechanistic investigations may be required in the future.

There are few studies on the ability of polysaccharide iron to act as an antioxidant in mammals, and therefore, we can only compare it with other types of organic iron. Dietary supplementation of 200 mg/kg lactoferrin [38] or 100 mg/kg glycine–iron complex [3] to sows increased serum antioxidant parameters in pregnant sows and newborn piglets. Similarly to these results, the present study showed that dietary TpFe supplementation significantly increased the activities of T-AOC, GSH-px, and CAT in cord blood and serum of weaned piglets, suggesting that maternal iron supplementation can improve offspring antioxidant capacity. The process may be explained by how tapioca polysaccharide binds ferric ions and inhibits the production of free radicals. Fe3+ chelated with polysaccharide does not break the basic structure of polysaccharide [39], and polysaccharide iron also has the basic structure of polysaccharide. Moreover, polysaccharide iron has more antioxidant activity than the original polysaccharide [40,41]. This is due to a change in the polysaccharide’s spatial structure after it was linked with iron, and the coordinating component interacts with the exposed active groups to promote free radical interaction, enhancing the ability to scavenge free radicals [42]. However, this study did not evaluate changes in the placental antioxidant capacity of sows or the liver antioxidant capacity of piglets since the focus of this study was the maternal dietary TpFe on offspring performance through cord blood; further research is needed in this regard.

In our study, we focused on changes in immunoglobulin IgA, IgM, and IgG and found that maternal supplementation with 100 mg/kg of TpFe increased IgG levels in sow blood, cord blood, and piglets. This is consistent with recent findings that IgG levels in piglet plasma were substantially related to survival, and serum IgG concentrations were lower in deceased piglets than in living piglets [43]. This also explained the differences in colostrum IgG found in the outcomes of the present research, indicating that maternal dietary supplementation with iron polysaccharide to enhance offspring performance involves the use of IgG as a crucial “bridge”.

Furthermore, RBCs play an essential role in tissue metabolism, and a sufficient number of RBCs are necessary to maintain tissue oxygenation and acid–base balance in the system [44]. Hb is a protein present in red blood cells that mainly distributes oxygen to the body’s tissues, and iron is an essential element for Hb [45]. Hb concentrations of 100 g/L or more show acceptable iron levels in an organism, while 80 g/L or higher is considered the anemia cutoff, and 60 g/L or higher is considered severe anemia [46]. HCT is the proportion of entire blood occupied by red blood cells and is affected by Hb content in RBC. The mean corpuscular volume is a good indicator of anemia after birth. In piglets that did not receive iron, the cells become microcytic, and the MCV will decrease [47]. Interestingly, compared to the inorganic-iron-fed sow group, the contents of RBC and Hb in cord blood and blood of piglets in the 100 mg/kg TpFe-supplemented sow group were significantly higher than the control group in our study. This was consistent with the findings of earlier research [20]. Moreover, the contents of HCT and MCV of piglets were significantly increased. These results match Hb levels and suggest that sows had greater iron status or that nursing piglets had increased iron bioavailability.

Interleukin-2 (IL-2) is a Th1 cytokine mainly produced by activated T lymphocytes that plays a regulatory role by activating and enhancing various functions of immune cells. It plays an important immuno-physiological role in the immune response system [15,48]. Interleukin (IL)-6 is a pleiotropic cytokine that plays essential functions in immunological response, inflammation, and hematopoiesis control [49]. The tumor necrosis factor alpha (TNF-α) is a cytokine with pleiotropic effects on numerous cell types. It has been identified as a significant regulator of inflammatory responses and is known to play a role in the etiology of various inflammatory and autoimmune diseases [50]. In a recent double-blind randomized trial examining the impact of iron supplementation in anemic pregnant women, iron treatment significantly raised maternal IL-2 levels compared to the control group [51]. In the present study, serum IL-2 levels in tapioca-iron-polysaccharide-fed sows significantly increased, although cord blood and piglet blood exhibited no change. Sow serum, cord blood, and piglet serum did not significantly differ in terms of IL-6 and TNF-. These results suggest that the effects of TpFe on serum inflammatory and immunological hormones were mainly maternal in nature and had little effect on passage.

Moreover, CEE and PRL can stimulate mammary gland growth and development, improve sow breastfeeding capacity, and improve reproductive success [52,53]. CCK is a typical brain and intestinal peptide that regulates the body to produce fullness signals and then inhibits eating activities in the digestive system and central and peripheral neurological systems [54]. GH is required for postnatal somatic growth and the retention of lean tissue in the animal at maturity. In the current study, there were no significant increases in serum estrogen levels in sow serum or cord blood, while the PRL concentration was considerably higher in serum and cord blood of sows given polysaccharide iron. Similarly, the serum CCK of sows was significantly higher than that of inorganic iron groups, which might explain the higher feed intake of nursing sows. Furthermore, GH levels were significantly increased in cord blood and piglet blood corresponding to the group of sows fed polysaccharide iron, indicating that maternal supplementation with TpFe had an innate developmental advantage in piglets.

Organic iron supply boosts pig birth weight and fetal iron reserves, decreases postnatal pig mortality rates and stillbirths, and leads to larger weaning weights of pigs, according to some earlier research [55]. Moreover, during the third trimester of pregnancy and lactation, sows fed an organic iron supplement (90 mg/kg) or an inorganic iron injection (60 and 80 mg/kg) were found to be more efficient, leading to a decrease in piglet mortality and higher weight at weaning [56]. These findings support our findings that sows given 100 mg/kg of TpFe had improved litter weight and average body weight at weaning and reduced piglet mortality. However, one study had different results that dietary chelated iron supplementation of 150 mg/kg to sows had no effect on piglet weight at weaning [57]. This may be related to the different types of iron chelates. In particular, the current study focused on a topic not covered in earlier research, the risk of diarrhea in suckling piglets, and found that maternal TpFe significantly reduced the probability of suckling piglets experiencing diarrhea. The cause of this may be the increased immunoglobulin levels in piglet serum and the observed better iron bioavailability of TpFe, which was consistent with the data discussed above.

The current data indicate a positive correlation between the live birth rate and the sows’ serum prolactin (PRL), which is consistent with earlier findings. An extremely strong and significant link between PRL and the number of live births (r = 0.670) suggested that improved litter size may be related to PRL [58]. Indeed, there is evidence that maternal oxytocin concentrations increase with litter size [59], possibly due to nest-building activity [60]. Urea nitrogen concentration is usually used to reflect the digestion of dietary protein. Thus, it was easy to understand the results of our experiment indicating that the diarrhea rate of suckling piglets was positively correlated with the urea nitrogen content in colostrum, as a high-protein diet is usually associated with causing diarrhea in piglets [61]. However, the level of colostrum IgM, IgG, and cord blood hemoglobin (Hb) was negatively correlated with the risk of diarrhea in suckling piglets. This is in agreement with the results of a previous study, which found a correlation between higher IgM and IgG group concentrations in sow colostrum and decreased incidence of diarrheal disease in newborn piglets [62]. The development of Hb in cord blood would undoubtedly strengthen piglets’ inbuilt immunity. Immune globulins, primarily IgG and IgM, are thought to be the main components of colostrum protein, and they shield piglets from possible harm caused by pathogenic bacteria [62].

Moreover, the mortality of suckling piglets was negatively correlated with the serum iron content of sows and IgM in colostrum, glutathione peroxidase (GSH-px), as well as IgG of cord serum. Although no study has directly focused on the connection between sows’ serum iron levels and piglet fatalities during lactation, it has been shown that sows fed organic iron had higher serum iron levels, which was followed by an increase in live litter size [2]. It is possible that iron-rich maternal nutrition enhances piglets’ ability to adjust throughout lactation and boosts the iron stored in the liver. Similarly, the congenital adaptability of the piglets was also increased, and the chance of dying while nursing was decreased by a rise in IgM in colostrum, GSH-px, and IgG in cord blood. Furthermore, there was a trend of increased litter weight at weaning and piglet single-head weight with the enhancement of IgM and IgG concentrations in colostrum and milk [63]. This was similar to the present study, where the average weight of weaning piglets at day 21 had a positive connection with colostrum IgM and IgG and cord serum red blood cell (RBC) count. This might be because the immunoglobulin system offers critical antimicrobial defense against a variety of diseases and can provide passive immunity to piglet development. Moreover, RBC and piglet weights revealed a similar relationship, as found in previous studies [64]. This might be because red blood cells are necessary for tissue metabolism, and more of them are required to support the tissue oxygenation required for piglets’ fast growth.

## 5. Conclusions

In conclusion, maternal supplementary 100 mg/kg TpFe can improve their feed intake and reproductive performance, adjust the nutritional composition of colostrum, enhance the antioxidant and immunological performance of piglets, and stimulate hormone release in blood, resulting in enhanced piglet production performance, compared with the same dose of inorganic iron. The Tpfe used in this experiment has obvious advantages. Additionally, maternal feeding of a half-dose (50 mg/kg) of TpFe, which can totally replace inorganic iron, appears in our findings. Furthermore, this study reveals the correlation between the components of colostrum, cord blood, and piglet performance when sows are fed supplements with TpFe. However, since this is the first investigation into feed supplementation with plant polysaccharide iron in pig models, there were limited references to previous studies. To confirm the mechanism, further animal experiments and studies involving genes related to placental iron transport in sows and organ iron storage in newborn piglets need to be conducted and further researched.

## Figures and Tables

**Figure 1 animals-13-02492-f001:**
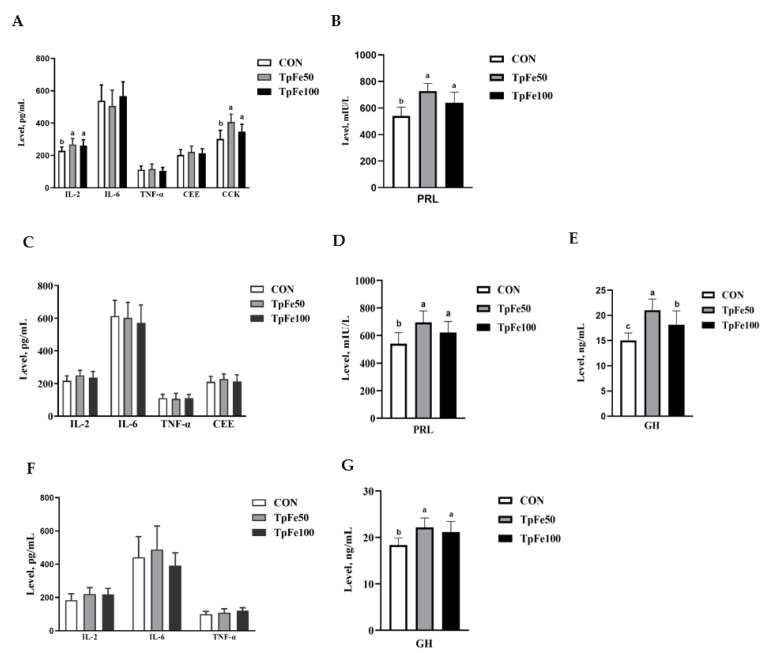
Bar chart of the effects of tapioca polysaccharide iron supplementation in sow diet on hormone levels in sow serum (**A**,**B**), cord blood serum (**C**–**E**), and piglet serum (**F**,**G**). The experiment included three treatments: CON group (basal diet added 100 mg/kg ferrous sulfate monohydrate), TpFe50 group (basal diet added 50 mg/kg tapioca polysaccharide iron), and TpFe100 group (basal diet added 100 mg/kg tapioca polysaccharide iron), *n* = 9 for each group. abc, different marked letters indicate significant difference (*p* < 0.05); IL-2, interleukin-2 (pg/mL); IL-6, interleukin-6 (pg/mL); TNF-α, tumor necrosis factor (pg/mL); CEE, estrogen (pg/mL); PRL, prolactin (mIU/L); CCK, cholecystokinin (pg/mL); GH, growth hormone (ng/mL).

**Figure 2 animals-13-02492-f002:**
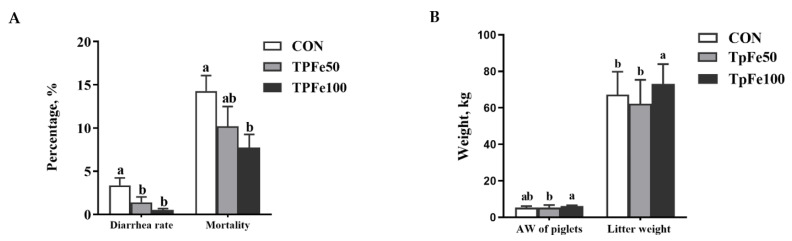
Bar chart of effects of tapioca polysaccharide iron supplementation in the sow’s diet on diarrhea rate, mortality of suckling piglets (**A**), and the average weight of individual weaned piglets and litter weight at age of 21 days (**B**). The experiment included three treatments: CON group (basal diet added 100 mg/kg ferrous sulfate monohydrate), TpFe50 group (basal diet added 50 mg/kg tapioca polysaccharide iron), and TpFe100 group (basal diet added 100 mg/kg tapioca polysaccharide iron), *n* = 20 for each group. ab, different marked letters indicate significant difference (*p* < 0.05); AW, average weight.

**Figure 3 animals-13-02492-f003:**
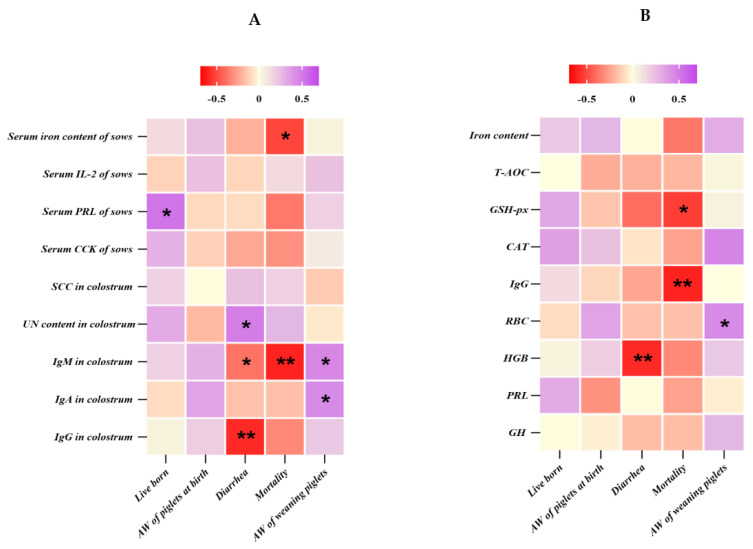
Heat map of Pearson correlation analysis between sow’s colostrum, serum active components and performance of newborn piglets (**A**); heat map of Pearson correlation analysis between cord blood and piglets serum active components and performance of newborn piglets (**B**). * *p* < 0.05; ** *p* < 0.01 (following the Pearson correlation analysis); IL−2, interleukin−2; PRL, prolactin; CCK, cholecystokinin; SCC, somatic cell count; UN, urea nitrogen; IgM, immune globulin M; IgA, immune globulin A; IgG, immune globulin G; AW, average weight; T−AOC, total antioxidant capacity; GSH−px, glutathione peroxidase; CAT, catalase; RBC, number of red blood cells; HGB, hemoglobin concentration; GH, growth hormone.

**Table 1 animals-13-02492-t001:** Basal diet composition and nutritional level of sows during late gestation and lactation (dry matter basis, %).

Item	Content	
	Gestation	Lactation
Ingredients		
Corn	24.95	40.00
Barley	20.00	12.30
Soybean meal, CP ≥ 43%	15.75	15.00
Brown rice	10.00	-
Wheat feed flour	2.00	14.00
Rice bran (defatted)	7.00	-
Sugar beet pulp	6.00	3.50
Yeast (brewers)	3.50	-
Puffing soybean	3.00	5.00
Puffing flaxseed	2.00	3.00
Fish meal, CP > 65%	-	1.00
Calcium hydrogen phosphate	1.40	1.40
Soybean oil	0.50	1.00
Sodium hydrogen carbonate	0.20	0.20
Zeolite powder	0.66	0.54
Salt	0.54	0.56
Premix ^1^	2.50	2.50
Total	100.00	100.00
Calculated composition		
Lysine (%)	0.95	1.03
Methionine (%)	0.37	0.40
Methionine + Cysteine (%)	0.63	0.70
Threonine (%)	0.68	0.74
Tryptophan (%)	0.19	0.23
Measured composition		
DE ^2^, kcal/kg	3205.06	3425.80
Crude protein (%)	15.56	16.50
Calcium (%)	0.89	0.89
Total phosphorus (%)	0.70	0.66
Iron (mg/kg)	30.19	32.41

^1^ The premix provides per kilogram of ration: vitamin A, 7000 IU; vitamin D3, 1800 IU; vitamin E, 160 IU; D-pantothenic acid, 16 mg; vitamin B_2_, 6 mg; folic acid, 6.8 mg; niacin, 70 mg; vitamin B_1_, 2 mg; vitamin B_6_, 3 mg; biotin, 0.75 mg; vitamin B_12_, 0.03 mg; copper (copper sulfate), 12.5 mg; zinc (zinc sulfate), 100 mg; manganese (manganese sulfate), 64 mg; iodine (calcium iodate), 0.6 mg; selenium (sodium selenite), 0.4 mg; chromium (chromium picolinate), 0.25 mg; sodium chloride, 5.4 g. ^2^ DE, digestive energy = gross energy in feed–gross energy in feces.

**Table 2 animals-13-02492-t002:** Effects of TpFe on feed intake of sows during late gestation and lactation ^1^.

Items	CON ^2^	TpFe50 ^3^	TpFe100 ^4^	SEM ^5^	*p*-Value
Gestational feed intake, kg/d	2.90	2.90	2.89	0.022	0.946
Lactation feed intake, kg/d	4.66 ^b^	4.56 ^b^	5.06 ^a^	0.070	0.007

^ab^ Means within a row with different superscripts differ (*p* < 0.05). ^1^ Data are shown as mean ± SEM (*n* = 20). ^2^ CON = basal diet + 100 mg/kg ferrous sulfate monohydrate. ^3^ TpFe50 = basal diet + 50 mg/kg tapioca polysaccharide iron. ^4^ TpFe100 = basal diet + 100 mg/kg tapioca polysaccharide iron. ^5^ SEM = standard error of the mean.

**Table 3 animals-13-02492-t003:** Effects of TpFe on reproductive performance of sows ^1^.

Item	CON ^2^	TpFe50 ^3^	TpFe100 ^4^	SEM ^5^	*p*-Value
Litter size, *n*	13.63	14.60	14.95	0.377	0.348
Number of live births, *n*	12.26 ^b^	13.45 ^ab^	14.65 ^a^	0.352	0.022
Stillbirths, *n*	1.15	1.15	1.10	0.195	0.993
Mummies, *n*	0.25	0.00	0.05	0.046	0.059
Duration of labor, min	270.17	343.42	287.64	16.431	0.188
Birth (alive) litter weight, kg	18.19 ^b^	18.09 ^b^	21.04 ^a^	0.503	0.021
Average weight of live-born piglets, kg	1.41 ^b^	1.38 ^b^	1.53 ^a^	0.023	0.017

^ab^ Means within a row with different superscripts differ (*p* < 0.05). ^1^ Data are shown as mean ± SEM (*n* = 20). ^2^ CON = basal diet + 100 mg/kg ferrous sulfate monohydrate. ^3^ TpFe50 = basal diet + 50 mg/kg tapioca polysaccharide iron. ^4^ TpFe100 = basal diet + 100 mg/kg tapioca polysaccharide iron. ^5^ SEM = standard error of the mean.

**Table 4 animals-13-02492-t004:** Effects of TpFe on active components in colostrum of sows ^1^.

Item	CON ^2^	TpFe50 ^3^	TpFe100 ^4^	SEM ^5^	*p*-Value
Total dry matter, %	17.88	18.21	18.63	1.358	0.977
Milk fat percentage, %	4.92	5.03	5.03	0.258	0.981
Milk protein percentage, %	16.78	17.84	18.31	1.005	0.415
Milk lactose percentage, %	2.63	2.37	2.93	0.119	0.158
Non-milk fat solid content, %	21.91	22.33	21.68	0.527	0.887
Somatic cell count, ×10^3^ cells/ml	633.09 ^a^	387.41 ^ab^	254.79 ^b^	57.600	0.022
Urea nitrogen content, mg/dL	48.96 ^a^	43.03 ^ab^	36.99 ^b^	1.956	0.043
Iron content, mg/L	4.42	4.86	4.98	0.405	0.871
IgM ^6^, g/L	1.21 ^c^	1.35 ^b^	1.67 ^a^	0.042	<0.001
IgA ^7^, g/L	0.52 ^b^	0.54 ^b^	0.69 ^a^	0.196	<0.001
IgG ^8^, g/L	10.59 ^b^	12.01 ^ab^	13.77 ^a^	0.441	0.016

^abc^ Means within a row with different superscripts differ (*p* < 0.05). ^1^ Data are shown as mean ± SEM (*n* = 9). ^2^ CON = basal diet + 100 mg/kg ferrous sulfate monohydrate. ^3^ TpFe50 = basal diet + 50 mg/kg tapioca polysaccharide iron. ^4^ TpFe100 = basal diet + 100 mg/kg tapioca polysaccharide iron. ^5^ SEM = standard error of the mean. ^6^ IgM, immunoglobulin M. ^7^ IgA, immunoglobulin A. ^8^ IgG, immunoglobulin G.

**Table 5 animals-13-02492-t005:** Effects of TpFe on the iron content of diet, sow serum, cord blood, and weaned piglet blood ^1^.

Item	CON ^2^	TpFe50 ^3^	TpFe100 ^4^	SEM ^5^	*p*-Value
Gestation diet, mg/kg	129.57 ^a^	80.52 ^b^	130.48 ^a^	10.453	<0.001
Lactation diet, mg/kg	133.06 ^a^	81.77 ^b^	132.39 ^a^	10.771	<0.001
Sow serum, mg/L	22.29 ^b^	25.18 ^b^	33.63 ^a^	1.739	0.011
Cord blood, mg/L	9.85 ^b^	13.43 ^ab^	17.02 ^a^	1.236	0.046
Weaned piglet blood, mg/L	16.43 ^b^	19.62 ^ab^	24.21 ^a^	1.239	0.030

^ab^ Means within a row with different superscripts differ (*p* < 0.05). ^1^ Data are shown as mean ± SEM (*n* = 9). ^2^ CON = basal diet + 100 mg/kg ferrous sulfate monohydrate. ^3^ TpFe50 = basal diet + 50 mg/kg tapioca polysaccharide iron. ^4^ TpFe100 = basal diet + 100 mg/kg tapioca polysaccharide iron. ^5^ SEM = standard error of the mean.

**Table 6 animals-13-02492-t006:** Effects of TpFe on the content of antioxidant active substances in serum of sows, umbilical cords, and piglets ^1^.

Item	Sample	CON ^2^	TpFe50 ^3^	TpFe100 ^4^	SEM ^5^	*p*-Value
MDA ^6^	sows	4.63	5.34	4.59	0.415	0.725
	cord	4.38	3.64	3.28	0.248	0.169
	piglets	4.77	5.27	3.24	0.368	0.112
T-AOC ^7^	sow	6.93	7.15	10.01	0.716	0.123
	cord	4.45 ^b^	6.67 ^a^	6.91 ^a^	0.425	0.033
	piglet	5.60 ^b^	5.86 ^b^	7.15 ^a^	0.205	0.001
GSH-px ^8^	sow	109.83	135.12	145.54	10.621	0.386
	cord	42.15 ^b^	50.95 ^b^	101.03 ^a^	7.507	<0.001
	piglet	104.01 ^b^	104.79 ^b^	142.37 ^a^	5.964	0.004
T-SOD ^9^	sow	30.93	35.87	36.40	1.292	0.156
	cord	30.17	31.03	31.38	1.684	0.959
	piglet	37.09	39.94	44.44	1.890	0.303
CAT ^10^	sow	11.38	13.79	13.92	0.771	0.300
	cord	9.29 ^b^	11.95 ^a^	12.53 ^a^	0.523	0.034
	piglet	10.40 ^b^	13.28 ^a^	13.49 ^a^	0.550	0.027

^ab^ Means within a row with different superscripts differ (*p* < 0.05). ^1^ Data are shown as mean ± SEM (*n* = 9). ^2^ CON = basal diet + 100 mg/kg ferrous sulfate monohydrate. ^3^ TpFe50 = basal diet + 50 mg/kg tapioca polysaccharide iron. ^4^ TpFe100 = basal diet + 100 mg/kg tapioca polysaccharide iron. ^5^ SEM = standard error of the mean. ^6^ MDA = malonaldehyde. ^7^ T-AOC = total antioxidant capacity. ^8^ GSH-px = glutathione peroxidase. ^9^ T-SOD = total superoxide dismutase. ^10^ CAT = catalase.

**Table 7 animals-13-02492-t007:** Effects of TpFe on immune and conventional substance active ingredient content of sow blood, cord, and piglet blood ^1^.

Item	Sample	CON ^2^	TpFe50 ^3^	TpFe100 ^4^	SEM ^5^	*p*-Value
IgM ^6^, g/L	sow	1.57	1.55	1.95	0.081	0.061
	cord	0.11	0.12	0.14	0.006	0.076
	piglet	0.70	0.81	0.84	0.063	0.641
IgA ^7^, g/L	sow	0.24	0.22	0.34	0.023	0.062
	cord	0.05	0.06	0.05	0.005	0.878
	piglet	0.11	0.14	0.15	0.009	0.261
IgG ^8^, g/L	sow	8.09 ^b^	8.52 ^ab^	8.77 ^a^	0.109	0.031
	cord	0.23 ^b^	0.36 ^a^	0.42 ^a^	0.026	0.005
	piglet	4.62 ^b^	5.98 ^b^	7.94 ^a^	0.449	0.003
RBC ^9^, ×10^12^/L	sow	5.73	5.74	5.87	0.105	0.838
	cord	4.15 ^b^	4.76 ^ab^	5.33 ^a^	0.209	0.042
	piglet	5.57 ^b^	5.63 ^b^	6.29 ^a^	0.114	0.006
Hb ^10^, g/L	sow	109.86	110.57	112.88	1.703	0.762
	cord	87.13 ^b^	96.00 ^ab^	102.57 ^a^	2.703	0.021
	piglet	79.50 ^b^	88.17 ^b^	104.17 ^a^	3.136	0.001
HCT ^11^, %	sow	33.81	32.90	33.24	0.606	0.836
	cord	30.87	34.42	32.82	0.673	0.144
	piglet	25.38 ^b^	28.98 ^ab^	33.27 ^a^	1.197	0.017
MCV ^12^, f L	sow	58.36	57.99	58.39	0.441	0.720
	cord	65.59	63.14	67.54	0.866	0.119
	piglet	46.00 ^b^	45.47 ^b^	54.66 ^a^	1.544	0.014

^ab^ Means within a row with different superscripts differ (*p* < 0.05). ^1^ Data are shown as mean ± SEM (*n* = 9). ^2^ CON = basal diet + 100 mg/kg ferrous sulfate monohydrate. ^3^ TpFe50 = basal diet + 50 mg/kg tapioca polysaccharide iron. ^4^ TpFe100 = basal diet + 100 mg/kg tapioca polysaccharide iron. ^5^ SEM = standard error of the mean. ^6^ IgM = immunoglobulin M. ^7^ IgA = immunoglobulin A. ^8^ IgG = immunoglobulin G. ^9^ RBC = number of red blood cells. ^10^ Hb = hemoglobin. ^11^ HCT = hematocrit. ^12^ MCV = mean corpuscular volume.

## Data Availability

The data presented in this study are available on request from the corresponding author.

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
