# Peer review of "Maternal Supplementary Tapioca Polysaccharide Iron Improves the Growth Performance of Piglets by Regulating the Active Components of Colostrum and Cord Blood"

_animals, 2023, doi:10.3390/ani13152492_

Round 1

Reviewer 1 Report

This article explored that maternal supplementation of tapioca polysaccharide iron improves the growth performance of piglets by regulating the active components of colostrum and cord blood

The experiment design was good and has certain guiding significance for the development of new organic iron supplements.

But some data showed confusing results and the article had some problems that should be considered.

Here are the specific suggestions.

1. The article repeatedly mentions the addition of TpFe during lactation, such as the simple summary and conclusion sections. However, TpFe supplement during lactation will not affect the active components of colostrum and cord blood of sows.

2. What are the units in the measured composition in Table 1? Such as iron.

3. Is the unit of feed intake for sow kg/d? If yes, please provide complete information. If not, please make corrections.

4. Please check the unit of Somatic cell count in Table 4.

5. In Figure 2B, does AW denote the average weight of piglets? Is the average weight of weaned piglets accurate?

6. The article raised the concern of modifying the litter size. Was the weight of the piglets' litter reevaluated after the litter size was adjusted? If not, the details presented in the article would be unreliable.

Minor editing of English language required

Author Response

Response to Reviewer 1 Comments

This article explored that maternal supplementation of tapioca polysaccharide iron improves the growth performance of piglets by regulating the active components of colostrum and cord blood

The experiment design was good and has certain guiding significance for the development of new organic iron supplements.

But some data showed confusing results and the article had some problems that should be considered.

Here are the specific suggestions.

Point 1: The article repeatedly mentions the addition of TpFe during lactation, such as the simple summary and conclusion sections. However, TpFe supplement during lactation will not affect the active components of colostrum and cord blood of sows.

Response 1: Dear reviewers,thanks to the reviewers for your high evaluation of this study.  I apologize if our expression gave you a bad impression. The manuscript's incorrect phrase "the addition of TpFe during lactation" has been changed to "maternal supplementary tapioca polysaccharide iron" and highlighted in simple summary and conclusion sections.

Point 2: What are the units in the measured composition in Table 1? Such as iron.

Response 2: The unit has been added to the item in Table 1 which measures the composition of the basal diet.

Point 3: Is the unit of feed intake for sow kg/d? If yes, please provide complete information. If not, please make corrections.

Response 3: Thanks for the reviewer's reminder, in fact, the unit of feed intake is "kg/d", we have modified it.

Point 4: Please check the unit of Somatic cell count in Table 4.

Response 4: The somatic unit was actually "×103 cells/ml," and we have corrected it.

Point 5: In Figure 2B, does AW denote the average weight of piglets? Is the average weight of weaned piglets accurate?

Response 5: Thank you very much for the suggestion. It is, in fact, the marking error in Figure 2B. Figure B's bar shape ought to state "AW of piglets" and "Litter weight" from left to right. We have modified the notes of the graphics.

Point 6: The article raised the concern of modifying the litter size. Was the weight of the piglets' litter reevaluated after the litter size was adjusted? If not, the details presented in the article would be unreliable.

Response 6: Dear reviewer, we apologize for your doubts caused by our unclear description of adjusting "litter size" in "2.3. Recording and Sample Collection". In fact, we adjusted the litter size among sows in the same group, resulting in a similar number of piglets per stall in the same group, so that the litter weights of the 20 sows in the same group were similar. We have described the adjustment of litter size with more detail in Section 2.3.

Look forward to hearing from you soon.

Kind regards,

Fang

Reviewer 2 Report

Report on the manuscript animals-2512864 entitled: Maternal supplementary tapioca polysaccharide iron improves the growth performance of piglets by regulating the active components of colostrum and cord blood.

-The authors have carried out a good discussion, but the description of the results (Results section) could be improved by considering the % of difference. Description such as “X was increased by Y” is poor. The results must be described as “X was increased a Y% when compared to ….

-In addition, the Conclusions must be rewritten. There is an interaction with TpFe dose. Some variables improved with both TpFe50 and TpFe100 and others only with TpFe100, therefore being dose dependent. Such observation has not been included in the Conclusions.  

-I have some concerns regarding the statistical analysis:

* L. 194-195, the authors say that “toward significance was noted at 0.05 < P < 0.10”. Nevertheless, there is not a single comment or description and discussion regarding such values. For example, “mummies” table 3, “IgM” table 7, etc.

* Figure 1 and 2: Please, explain the meaning of the “error bars”. Are they SD?
Why is the full bar displayed in some cases and only half of it in others?

In addition, considering the same “n” and the error bars that are shown, some discrepancies can be observed. For example, Fig 1B, please review the test for mean separation, with such error bars a statistical difference should be detected, etc…

-Figure 3 and L. 330-342. Neither the usefulness nor the relevance of this figure is clear.

In fact, the authors have considered a correlation of 0.5 as a limit. However, this value has no impact without the development of some regression or estimation model.

Also, please review Fig 3b CAT because either something is missing, or it does not add up.

Specific comments:

-          L. 27 and 110. Another “basal” diet?

-          L. 43. “adversely”? meaning?

-          L. 65 (62-65). “more effective”, meaning? What for?

-          L. 73-79. The authors talk about a “macromolecular polysaccharide = polysaccharide-iron complex” (references 10 and 11).
Afterwards, the authors talk about the new iron complex generated from tapioca starch.
Is it the same iron complex? If not, what is the relationship between the compound from references 10 and 11 and the new compound?

-          L. 115. “sq m”? meaning m2?

-          L. 155. Reference format?

-          L. 199. Similar?

--

Author Response

Response to Reviewer 2 Comments

Report on the manuscript animals-2512864 entitled: Maternal supplementary tapioca polysaccharide iron improves the growth performance of piglets by regulating the active components of colostrum and cord blood.

Point 1: The authors have carried out a good discussion, but the description of the results (Results section) could be improved by considering the % of difference. Description such as “X was increased by Y” is poor. The results must be described as “X was increased a Y% when compared to ….

Response 1: Dear reviewers,thanks to the reviewers for your professional and authoritative advice on this study.  I apologize if our expression for the “result section” gave you a bad impression. We followed your advice and modified the complete manuscript's results section. Highlights are employed.

Point 2: In addition, the Conclusions must be rewritten. There is an interaction with TpFe dose. Some variables improved with both TpFe50 and TpFe100 and others only with TpFe100, therefore being dose-dependent. Such observation has not been included in the Conclusions.

Response 2: With the guidance of the experts' suggestions, we rewrote the conclusion to more accurately represent the research purpose mentioned in the manuscript's prologue.

Point 3: I have some concerns regarding the statistical analysis:

* L. 194-195, the authors say that “toward significance was noted at 0.05 < P < 0.10”. Nevertheless, there is not a single comment or description and discussion regarding such values. For example, “mummies” table 3, “IgM” table 7, etc.

* Figure 1 and 2: Please, explain the meaning of the “error bars”. Are they SD?
Why is the full bar displayed in some cases and only half of it in others?

In addition, considering the same “n” and the error bars that are shown, some discrepancies can be observed. For example, Fig 1B, please review the test for mean separation, with such error bars a statistical difference should be detected, etc…

Response 3: * After reviewing the data statistics, we determined that this test was not a concentration gradient test, and the improper description of sentences L194-195 was removed with no consideration for "toward significance."

* The “error bars” of the Figure 1 and 2 means SD, and we apologized for the errors in the software draw that led to the discrepancies, which we have already fixed the bar chart. In addition, there was a significant difference in serum PRL between the three groups of sows, as mentioned in the results section; we apologize for presenting inaccurate information in the graph; and the bar chart has been redone.

Point 4: -Figure 3 and L. 330-342. Neither the usefulness nor the relevance of this figure is clear.

  In fact, the authors have considered a correlation of 0.5 as a limit. However, this value has no impact without the development of some regression or estimation model.

Also, please review Fig 3b CAT because either something is missing, or it does not add up.

Response 4: Dear reviewer, we reviewed Figure 3 and L. 330-342 and realized that neither its usefulness nor relevance of this figure is clear, because of our unclear comments on Figure 3. As a result, we modified the notes in Figure 3 and added the correlation coefficient r value in the SPSS Pearson Correlation analysis prediction model in the results part of L.330-342 to guarantee that the p-value we obtained was correct. What’s more, we reviewed the CAT data in Figure 3B. Because the heat map between them is dark in color, I assume you believe there is a positive correlation between CAT and AW of weaning piglets. However, the SPSS analysis findings show that P=0.067 and r=0.407 are almost significant. This may make the heat map look darker, but in fact, the P-value is in excess of 0.05. Therefore, no something is missing, or it does not add up (such as *) here, thanks for your rigorous thought.

Point 5: Specific comments:

-          L. 27 and 110. Another “basal” diet?

-          L. 43. “adversely”? meaning?

-          L. 65 (62-65). “more effective”, meaning? What for?

-          L. 73-79. The authors talk about a “macromolecular polysaccharide = polysaccharide-iron complex” (references 10 and 11).
  Afterwards, the authors talk about the new iron complex generated from tapioca starch.
Is it the same iron complex? If not, what is the relationship between the compound from references 10 and 11 and the new compound?

-          L. 115. “sq m”? meaning m2?

-          L. 155. Reference format?

-          L. 199. Similar?

Response 5:  -L. 27 and 110. Another “basal” diet?  The phrase "a Basal diet" has been changed to "the Basal diet," as there is no other basic diet, and this particularly refers to the "Basal diet" that we described.

- L. 43. “adversely”? meaning?     Replace it with "negatively" to avoid ambiguity.

- L. 65 (62-65). “more effective”, meaning? What for?  The original sentence has been altered with "Numerous studies on the efficacy of iron supplementation in pregnant sows have demonstrated that organic iron supplements, such as ferrous fumarate and amino acid-chelated iron, which have been confirmed to perform better than inorganic iron supplements in animal production applications", since the grammar of the language was not good.

 - L. 73-79. The authors talk about a “macromolecular polysaccharide = polysaccharide-iron complex” (references 10 and 11).  Afterwards, the authors talk about the new iron complex generated from tapioca starch. Is it the same iron complex? If not, what is the relationship between the compound from references 10 and 11 and the new compound?  For clarity, we have reorganized the sentence “saccharide-iron complexes as novel iron supplements have aroused attention for the high iron absorption rate and no gastrointestinal irritation at oral doses. In addition, research on the biological activities of saccharide-iron complexes revealed that they also exhibited good abilities in treating anemia, eliminating free radicals, and regulating the immune response.” So that we are in fact talking about complexes of sugar and iron, not about macromolecular polysaccharides. To better illustrate this point, we have replaced the previous literature [10] with the latest polysaccharide iron literature [11]." Then we proposed that the type of polysaccharide iron used in this experiment is a new polysaccharide iron complex formed by polysaccharide extracted from cassava starch and ferric chloride. Therefore, cassava polysaccharide iron used in this test belongs to a class of polysaccharide iron complex products, but such products have not been reported.
   - L. 115. “sq m”? meaning m2? - L. 155. Reference format?  Yes, it is, to make it easier for the reader to understand, we changed it to "m2 ".

- L. 199. Similar?  We changed P=0.007 to P < 0.05 according to author's guide.

Look forward to hearing from you soon.

Kind regards,

Fang

Round 2

Reviewer 2 Report

The authors have taken into account the reviewer's comments and the manuscript has been improved.

Some typos.